

# Stochastic field dynamics in models of spontaneous unitarity violation

Lotte Mertens[1,2], Matthijs Wesseling[1] and Jasper van Wezel[1⋆]

**1** Institute for Theoretical Physics Amsterdam, University of Amsterdam,
Science Park 904, 1098 XH Amsterdam, The Netherlands
**2** Institute for Theoretical Solid State Physics, IFW Dresden,
Helmholtzstr. 20, 01069 Dresden, Germany

⋆ vanwezel@uva.nl

## Abstract

**Objective collapse theories propose a solution to the quantum measurement problem by predicting deviations from Schrödinger's equation that can be tested experimentally. A class of objective theories based on spontaneous unitarity violation was recently introduced, in which the stochastic field required for obtaining Born's rule does not depend on the state of the system being measured. Here, we classify possible models for the stochastic field dynamics in theories of spontaneous unitarity violation. We show that for correlated stochastic dynamics, the field must be defined on a closed manifold. In two or more dimensions, it is then always possible to find stochastic dynamics yielding Born's rule, independent of the state being measured or the correlation time of the stochastic field. We show that the models defined this way are all isomorphic to the definition on a two-sphere, which we propose to be a minimal physical model for the stochastic field in models of spontaneous unitarity violation.**

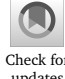

# 1   Introduction

Quantum mechanics has been proven to be highly accurate in describing the dynamics of microscopic systems [1, 2]. The dynamics of macroscopic objects as a whole, however, is described by classical physics. The qualitative distinction between these regimes leads to a problem that becomes visible when measuring microscopic objects using macroscopic measurement machines [3–6]. Even though the measurement machine consists of quantum particles, we cannot describe its observed behaviour using quantum mechanics. Precisely what causes the quantum-classical crossover and how the measurement device turns a superposed initial state into a single classical outcome is still unknown. These questions are collectively known as the quantum measurement problem, and it continues to be a topic of active research [1, 7–11]. Besides the formulation of explicit models for the quantum-classical crossover [7, 8, 12, 13], there has recently been a focus on constructing new experiments aimed at probing the crossover regime [9–11, 14, 15].

Theoretical approaches to solving the quantum measurement problem can broadly be divided into interpretations and objective collapse theories. Interpretations assume that Schrödinger's equation holds at all scales and then attempt to explain why we only perceive single classical states for macroscopic objects [1]. On the other hand, objective collapse theories –which are the focus of the current work– propose objectively distinct dynamics for the microscopic and macroscopic worlds, connected by a smooth transition in the mesoscopic region [7, 12, 16]. The transition is facilitated by a modification to Schrödinger's equation that leads to the "quantum state reduction" or collapse of superpositions into classical states. Although the modified dynamics applies to all objects, its effect is instantaneous for macroscopic objects but takes an almost infinitely long time to become significant in microscopic systems. At mesoscopic scales, it must then occur within measurable times. Currently, this mesoscopic regime is on the verge of being probed by state-of-the-art experimental efforts [1, 14, 15, 17–23].

All viable objective collapse theories must reproduce known experimental results correctly. This implies that their time evolution must include at least a stochastic, non-unitary term as well as a non-unitary and non-linear element [8, 24], because a fully deterministic theory cannot yield probabilistic measurement outcomes, while a fully linear theory cannot reproduce the observed stability and statistics of measurement outcomes. The non-unitary and non-linear element is thus required to introduce stable fixed points in the dynamics, while the combination of non-unitary stochastic and non-linear terms allows for the correct statistics to be realized [24]. All objective collapse theories moreover predict an instantaneous reduction of quantum superpositions into a classical measurement outcome and ensure Born's rule in the macroscopic limit [25, 26]. Despite this similarity in behaviour at the macro-scale, predictions of objective collapse theories may differ wildly in the mesoscopic region, and this provides opportunity for experimentally distinguishing between them [17, 18, 27].

Mathematically, one major distinction between different objective collapse theories is how stochasticity is introduced in their dynamics. In many models, the time evolution of the quantum state itself is assumed to be stochastic [1, 12, 13, 28]. Constructing the theory this way al-

lows for the explicit exclusion of faster-than-light communication and the automatic obtaining of Born's rule. It comes at the cost, however, of the stochastic contribution not being independent of the quantum state being measured [29]. It was recently shown that it is also possible to construct models in which the stochastic and non-linear contributions to the dynamics are decoupled, so that the stochastic term may represent a physical field that is independent of the state being measured [30]. These Spontaneous Unitarity Violation (SUV) models still allow for obtaining Born's rule [24, 30]. The exclusion of faster-than-light communication has not been demonstrated yet and is an important consideration for future work. However, it has been shown that this is possible for certain non-linear models [31].

Here, we derive constraints on the form of the stochastic terms that can appear in SUV models based on the requirement that the late-time probability distribution of measurement outcomes adheres to Born's rule. Since the stochastic term represents a physical field in these theories, identifying its possible symmetries and dimensionality narrows the possibilities for its physical origin.

In this study, we restrict attention solely to quantum state reduction starting from an initial superposition of two states. Notice that any successful objective collapse theory must be able to describe the quantum state reduction of an initial superposition over two pointer states (distinguished, for example, by their centre of mass positions). We focus on these processes because all possible SUV models reduce to one of only two forms when applied to the evolution of a two-state system. We thus identify general constraints on the stochastic field appearing in any theory of SUV. More stringent constraints on the stochastic field of specific models may perhaps be obtained by considering different initial conditions, which we leave for future research. We find that the requirement of Born's rule being obtained in the presence of correlated stochasticity and regardless of system size, requires a unique form of the two-state SUV dynamics, and we fully constrain the stochastic field parameters appearing in that form.

## 2 Two-state collapse dynamics

Consider the time evolution of the general two-component wave function parameterised on the Bloch sphere:

$$|\psi_t\rangle = \cos(\theta_t/2)|0\rangle + \sin(\theta_t/2)|1\rangle . \tag{1}$$

Here, the states $|0\rangle$ and $|1\rangle$ represent entangled (product) states of a microscopic quantum system in an eigenstate of the observable being measured and a macroscopic measurement device indicating the corresponding measurement outcome. In the generic description of a strong measurement [25], this is the state obtained right after coupling pointer states of a measurement machine to the quantum system being measured. Collapse, or measurement, should reduce the system to either $|0\rangle$ or $|1\rangle$ with respective probabilities $\cos^2(\theta_0/2)$ and $\sin^2(\theta_0/2)$. This quantum states reduction should take place over immeasurably short times for truly macroscopic measurement machines, while taking indefinitely long times if the measurement device is made microscopic. Notice that we ignored the relative phase between between components of $|\psi_t\rangle$, which is generically present, as well as the dynamics of the overall norm and phase. As has been previously shown [24, 32, 33], these do not influence the time evolution of $\theta_t$, even for general (not necessarily unitary) models of quantum state evolution.

Here, we focus on models of Spontaneous Unitarity Violation (SUV), in which the stochastic contribution to the state dynamics is driven by a physical field evolving independently from the quantum state [29]. Within this setting, imposing that there are only two fixed points in the dynamics (i.e. two possible measurement outcomes), and that the probabilities of reaching either fixed point correspond to Born's rule (i.e. are given by the squared amplitude of the corresponding component in the initial wave fuction) severely constrains the possible form

of the evolution. In fact, up to unitary transformations, the only allowed forms for the non-unitary evolution of a two-state system are [24, 30]:

$$\dot{\theta}_t = -JN\sin(\theta_t)(\cos(\theta_t + a_1) - \lambda_1(t)), \tag{2}$$

$$\text{or} \quad \dot{\theta}_t = -JN\sin(\theta_t)\cos(\theta_t + a_2 - \lambda_2(t)). \tag{3}$$

Here $N$ indicates the system size (the volume, mass, or number of constituent particles of the measurement machine), which explicitly indicates the origin of the collapse dynamics stemming from a process of spontaneous symmetry breaking involving states with distinct values of an emergent order parameter [8, 29, 34]. The coupling constant $J$ determines the speed of the collapse process, and $\theta$ represents the angle with the $z$-axis on the Bloch sphere as before. The variables $\lambda_{1,2}(t)$ and $a_{1,2}$ denote time-dependent stochastic variables and constant parameters respectively. The allowed probability distribution functions (pdf) and dynamics of $\lambda_{1,2}$ and values of $a_{1,2}$ are constrained by the requirement that there is no inherent preference (independent of the initial state) for either the $\theta = 0$ or the $\theta = \pi$ measurement outcome. This means that $\dot{\theta}(\theta, \lambda) = -\dot{\theta}(\pi - \theta, \lambda')$ for some value $\lambda'$ appearing with the same probability as $\lambda$. We thus find that necessarily $a_1 = 0$ and $\lambda_1$ has a probability distribution function that is even, while the pdf of $\lambda_2$ must be symmetric around $\lambda_2 = a_2$. Without loss of generality, we can then also consider an even pdf for $\lambda_2$ and set $a_2$ to be zero.

Notice that Eqs. (2) and (3) neglect the usual unitary part of the evolution described by the quantum mechanical Hamiltonian. As shown previously [24], the statistics of measurement outcomes is not influenced by the unitary part of the dynamics and we thus set it to zero without loss of generality. Moreover, by studying the dynamics directly on the Bloch sphere, we avoid the need for normalisation of the wave function. Notice that normalisation can be included at the level of the state dynamics [29], but that this leaves Eqs. (2) and (3) invariant. These thus represent the only possible forms of the quantum state reduction process consistent with having stable end states corresponding to single pointer states (the states $|0\rangle$ and $|1\rangle$), having no possible other end states (no attractive fixed points other than $\theta = 0$ or $\theta = \pi$), and containing an independent physical field driving the stochastic evolution (i.e. $\lambda_{1,2}$ evolving independently of $\theta$, and not being multiplied by any $\theta$-dependent factor besides the overall, geometric factor $\sin(\theta)$ which constrains the evolution to the Bloch sphere) [24, 29, 30].

With either of the equations (2) or (3), the probability of obtaining a particular outcome in any given realisation of the dynamics depends on the probability distribution and dynamics of the stochastic variable $\lambda_{1,2}(t)$. Assuming this variable arises from an as-yet unknown physical process, its effectively stochastic dynamics is characterised by a correlation time $\tau$, which cannot be identically zero. The average half-time $\tau_c$ of the quantum state reduction towards the poles of the Bloch sphere, meanwhile, is determined by the overall factor $NJ$, rendering it inversely proportional to system size. The ratio of these two intrinsic time scales defines three possible regimes for the collapse dynamics.

- Macroscopic regime: $\tau_c \ll \tau$. The term $\lambda$ is approximately constant during quantum state reduction.

- Mesoscopic regime: $\tau_c \sim \tau$.

- Microscopic regime: $\tau_c \gg \tau$. The term $\lambda$ fluctuates strongly during collapse.

Here, the names of the regimes refer to the size $N \propto 1/\tau_c$ of the measurement device. Effectively instantaneous collapse into classical measurement outcomes, as envisioned for devices used in everyday quantum measurement, occurs in the macroscopic regime. In this regime, the collapse dynamics is much faster than any other time scale, including that characterising the stochastic variable, and $\lambda$ may be approximated to be constant. In contrast, if both the

object to be measured and the "measurement device" are microscopic quantum systems, we are in the microscopic regime and all evolution should be extremely well-approximated by the unitary Schrödinger equation. The value of $J$ should thus be such that the collapse time in this regime is larger than any observable time scale. The third, mesoscopic, regime interpolates between these extremes. It has not been probed in any experiments to date [9–11, 14, 15]. It is the regime where new physics might be found, and where objective collapse theories can be distinguished from one another.

## 3 Macroscopic regime

The dynamics of Eqs. (2) and (3) must reproduce existing experiments, and thus yield measurement outcomes adhering to Born's rule in the macroscopic regime. That is, an ensemble of infinitely many evolutions starting from the same initial state but being propagated according to Eq. (2) or (3) using different $\lambda_{1,2}$, should only contain evolutions ending up in either $|0\rangle$ or $|1\rangle$, and the proportion of evolutions reaching $|0\rangle$ should be $\cos^2(\theta_0/2)$.

Assuming the stochastic variable $\lambda_{1,2}$ to be constant during the nearly instantaneous collapse dynamics in the macroscopic regime, its value is taken from an equilibrium probability distribution. If the initial value of $\cos(\theta_0)$ is larger than $\lambda_1$ in Eq. (2), the derivative $\dot{\theta}$ will be negative. The value of $\theta_t$ will then decrease over time, while the value of $\cos(\theta_t)$ increases and $\dot{\theta}$ becomes even more negative. This continues until the value $\theta = 0$ is reached (asymptotically), which indicates the completion of the collapse process and realisation of a single classical pointer state. As the initial parameter values completely determine the measurement outcome, the probability of obtaining the outcome $|0\rangle$ in any individual quantum measurement can be written as:

$$P_1^{|0\rangle}(\theta_0) = \int_{l_1}^{h_1} \rho_{\text{eql},1}(\lambda)\Theta(\cos(\theta_0) - \lambda)\,d\lambda. \tag{4}$$

Here, $\Theta$ is the Heaviside step function, while $\rho_{\text{eql},1}(\lambda)$ is the equilibrium probability distribution function for $\lambda_1$, defined on the domain $[l_1, h_1]$.

Starting instead from Eq. (3), we can make the same argument of the state evolving uniformly towards either $\theta = 0$ or $\theta = \pi$ for any given initial value of $\cos(\theta_0 - \lambda_2)$ as long as $\lambda_2 \in [-\pi/2, \pi/2]$. Assuming this, and using the Heavyside step function to restrain the limits on the integral, we thus write:

$$P_1^{|0\rangle}(\theta_0) = \int_{l_1}^{\cos(\theta_0)} \rho_{\text{eql},1}(\lambda)d\lambda, \tag{5}$$

$$P_2^{|0\rangle}(\theta_0) = \int_{\theta_0 - \pi/2}^{h_2} \rho_{\text{eql},2}(\lambda)d\lambda. \tag{6}$$

Imposing $P_{1,2}^{|0\rangle}$ to vanish as $\theta_0$ goes to $\pi$, fixes the remaining limits on the integrals to be $l_1 = -1$ and $h_2 = \pi/2$. Considering the probabilities for measuring $|1\rangle$ similarly fixes $h_1 = 1$ and $l_2 = -\pi/2$. Substituting those limits and demanding that the probabilities in Eqs. (5) and (6) equal Born's rule for general $\theta_0$ yields the necessary forms for the probability distribution functions:

$$\rho_{\text{eql},1}(\lambda) = \frac{\sin(\arccos(\lambda))}{2\sqrt{1 - \lambda^2}} = \frac{1}{2}, \tag{7}$$

$$\rho_{\text{eql},2}(\lambda) = \frac{1}{2}\sin(\pi/2 - \lambda). \tag{8}$$

We thus finally find the SUV dynamics for two-component superpositions to be described by:

$$\dot{\theta} = -JN \sin(\theta)(\cos(\theta) - \lambda_1), \tag{9}$$

$$\text{or} \quad \dot{\theta} = -JN \sin(\theta) \cos(\theta - \lambda_2), \tag{10}$$

with the pdf of the stochastic variables $\lambda_{1,2}$ given in the limit of infinite correlation time by a normalised flat distribution between $-1$ and $1$ for $\lambda_1$, and by $1/2 \cos(\lambda)$ on the domain $[-\pi/2, \pi/2]$ for $\lambda_2$. Notice these are still just Eqs. (2) and (3) with the parameters $a_{1,2}$ set to zero and the forms of the pdf for the stochastic variables $\lambda_{1,2}$ specified.

For both of the possible evolution equations, the stochastic variable is necessarily defined on a bounded domain. If the stochastic variable represents a physical process, this implies that the process must be defined on a bounded domain as well. It is then most natural to consider processes on closed (periodic) manifolds, as opposed to open manifolds with arbitrary boundary conditions. Moreover, although the stochastic variable is essentially constant during collapse in the macroscopic regime, it must necessarily fluctuate between measurements in order to allow for different measurement outcomes being realised in subsequent measurements. The stochastic variable must therefore evolve (randomly) in time, and have a finite correlation time. Finally, to ensure that subsequent measurements are independent of one another, the future evolution of the stochastic variable should not depend on its past values. Taking the limit in which the variable has no memory of its past at all, we consider the stochastic process to be Markovian.

Combining these requirements, the stochastic variable can be interpreted as the (abstract) position of a (Markovian) random walk with correlation time $\tau$ on a closed manifold. The shape of the manifold on which the walk takes place fully determines the probability distribution function for its position. In the following sections, we consider which manifolds allow for random walks that realise the pdf required for obtaining Born's rule.

## Random walk on a circle

The lowest-dimensional closed manifold to consider is the circle, with an angle $\eta$ parameterising its single dimension. We will not consider more general shapes, since smooth deformations of the circle and of the pdf for the coordinate $\eta$ can always be made to cancel one another, making all smoothly connected shapes equivalent.

For times short compared to the correlation time $\tau$, a Markovian random walk will result in a Gaussian pdf centered around the initial value of the angle. For times (infinitely) larger than $\tau$, the pdf $\rho_{\text{eql,circ}}(\eta)$ becomes flat, indicating that each angle on the circle is equally likely to be realised. This is a general feature of Markovian random walks on a closed manifold; as there is no preferred point, each point on the manifold will become equally likely in the infinite time limit.

To see whether the coordinate $\eta$ can be used to define the random variables $\lambda_{1,2}$ yielding Born's rule, we need to identify a function $\lambda(\eta)$ such that the pdf for $\lambda$ equals that of Eq. (7) or (8). The pdf for the value of any function $\lambda(\eta)$ is related to the pdf for its argument $\eta$ through [35]

$$\rho(\lambda(\eta))|d\lambda/d\eta| = \rho(\eta). \tag{11}$$

Because the coordinate $\eta$ is periodic, it is defined only modulo integer multiples of $2\pi$ (its period). For $\lambda_1(\eta)$ to be invariant under additions of $2\pi$, it necessarily needs to be a trigonometric function, such as $\lambda_1 = \cos(\eta)$, $\lambda_1 = \sin(\eta)$, or polynomial combinations of these. Using $\rho_{\text{eql,circ}}(\eta) = 1/(2\pi)$ and Eq. (11), none of these trigonometric functions can produce the pdf of Eq. (7). Similarly, because $\lambda_2$ appears inside a cosine in Eq. (10), it should be an angle, defined modulo $2\pi$. This is realised if $\lambda_2$ is a linear function of $\eta$, but it is not possible to obtain the pdf of Eq. (8) that way.

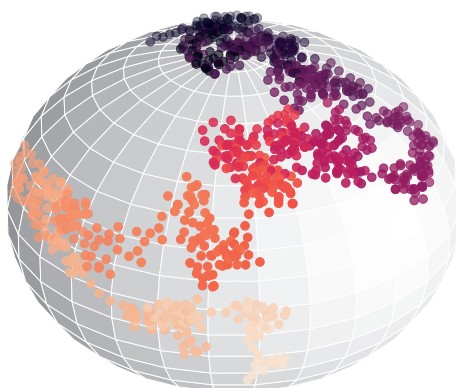

Figure 1: Typical random walk on the unit sphere with initial position $\eta_0 = 0$ and $\phi_0 = \pi/2$. One thousand steps were taken with an arc length of 0.06. The colours indicate the time evolution, going from dark to light.

We thus find that a Markovian random walk on the circle cannot generate the stochastic variable required for obtaining Born's rule. Additional degrees of freedom are required, which can be provided by considering higher dimensional manifolds.

**Random walk on a sphere**

The surface of a sphere is the representative closed manifold to be considered in two dimensions. We will discuss topologically distinct manifold such as the torus in Sec. 4. To parameterise the position of a random walk on the sphere we use the latitude and azimuth angles $\eta$ and $\phi$. In the limit of times (infinitely) long compared to the correlation time $\tau$, the probability distribution function for the random walk uniformly covers the sphere. This implies $\rho_{\text{eql,sphere}}(\phi) = 1/(2\pi)$ and $\rho_{\text{eql,sphere}}(\eta) = \sin(\eta)/2$ (see Supplemental Material for details).

As before, $\lambda_1$ should be a trigonometric function of the angles $\eta$ and $\phi$, while $\lambda_2$ is a linear function of them. Choosing the specific forms $\lambda_1 = \cos(\eta)$ and $\lambda_2 = \eta$, the uniform distribution of the random walk over the sphere combined with Eq.(11) results precisely in the probability distribution functions of Eqs. (7) and (8).

A Markovian random walk on the sphere can therefore be used to generate stochastic variables with the pdf required for obtaining Born's rule in the macroscopic regime. We will comment on random walks in higher dimensions and on topologically distinct manifolds in Sec. 4, after considering the effect of the random walk if the collapse time is comparable to the correlation time.

## 4 Mesoscopic regime

If the collapse time is finite and larger than the correlation time, $\tau_c > \tau$, the value of the stochastic variable will fluctuate significantly while the system collapses from a superposed state to a classical outcome. Experimental verification for the statistics of obtained outcomes is unavailable throughout a large part of this regime. It is easily verified, however, that any deviation of the late-time probabilities from Born's rule would allow faster-than-light communication, because if two observers share a known entangled state and a prior agreement of who measures first, the latter observations reproduce the measurement statistics of the earlier ones. We therefore require Born's rule also in this regime.

We again take the random walk on a sphere as the process defining the evolution of the stochastic variable. To find its probability distribution given a previous value, we use the fact that it must be Gaussian in the distance travelled. Starting from the point $\eta_0 = 0$, the naive expectation is then for the probability distribution to spreads over time as:

$$\rho(\eta, \phi) \propto \exp\left[-\eta^2/(2\sigma^2)\right].$$

Here, $\eta$ equals the arc distance from the pole on a unit sphere for a point with latitude coordinate $\eta$. The variance of the distribution grows over time and is given by $\sigma^2(t) = Dt = \epsilon^2 t/\delta t$, with $\epsilon$ the typical arc distance travelled in time $\delta t$ so that $D$ the diffusion coefficient for the spread of the probability distribution, which is inversely proportional to its correlation time.

As before, only the probability distribution for the latitude angle $\eta$ affects the time evolution of the state. It is found by integrating over the azimuthal distribution:

$$\rho(\eta) \propto \int_0^{2\pi} d\phi \, \sin(\eta)\rho(\eta, \phi).$$

Here, $\sin(\eta)d\phi$ is the arc distance along a line of constant latitude.
Taking $\rho(\eta, \phi) \propto \exp\left[-\eta^2/(2\sigma^2)\right]$ as before, gives a good approximation for the distribution function at short times, but it fails to account for any random walks crossing the south pole. To correct for this, we can extend the distribution beyond $\eta = \pi$ and fold it back onto the unit sphere:

$$\rho_A(\eta) = \bar{N} \sum_{n=-\infty}^{\infty} \int_0^{2\pi} d\phi \, \sin(\eta)e^{-\frac{1}{2\sigma^2}(\eta+n2\pi)^2}. \tag{12}$$

Here, $n \in \mathcal{Z}$ and $\bar{N}$ is a normalisation factor. Finally, starting from an arbitrary initial position on the sphere the same sequence of arguments yields the expression:

$$P_A(\eta) = \sum_{n=-\infty}^{\infty} \int_0^{2\pi} d\phi \bar{N} \sin(\eta) \exp\left(\frac{-1}{2\sigma^2}(\arccos[\sin(\eta_0)\sin(\eta)\cos(\phi_0 - \phi)\right.$$
$$\left. + \cos(\eta_0)\cos(\eta)] + 2n\pi)^2\right). \tag{13}$$

Here, the arccos term is the arc distance between the initial point $(\eta_0, \phi_0)$ and the point $(\eta, \phi)$. We thus arrive at an exact but open form expression for the probability distribution obtained in a random walk on the sphere. Cutting off the sum at a finite value of $|n|$, it can be used to numerically compute the time evolution of Eqs. (2) and (3) in the mesoscopic regime.

**Numerical results**

Besides using an approximate form for the probability distribution function after a given time interval, as provided by Eq. (13), we can also numerically simulate the random walk on the unit sphere directly. Every time step, the stochastic variable then travels a fixed arc distance $d_{\text{step}}$ in a random direction. A typical trajectory on the unit sphere is displayed in Fig. 1
The statistical distribution of latitude angles after a given time and starting from a given position will approach the probability distribution function $\rho(\eta)$ when averaged over sufficiently many instances of the random walk. In Fig. 2, the evolution of the distribution is shown for different times. The Ansatz of equation 13 up to $|n|_{\text{max}} = 500$ is drawn as black lines in the same figure. here we used the definition $\sigma^2 = d_{\text{step}}^2 t/2$. Increasing $n_{\text{max}}$ yields an increasingly better match with the numerical averages. The $n = 0$ term on its own provides the approximation $\rho(\eta) \propto \sin(\eta)\exp\left[-(\eta - \eta_0)^2/(2\sigma^2)\right]$, which is indicated by the orange line. It gives a good approximation only at short times, and increasingly diverges from the numerical average as time progresses.

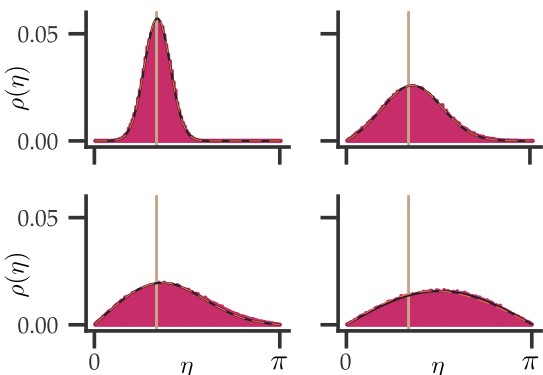

Figure 2: Probability distribution for the latitude angle $\eta$, after a random walk on the sphere with 10, 500, 1000, and 10000 steps of arc length 0.03. Here, we used the initial values $\eta_0 = \pi/3$ (indicated by a vertical light brown line) and $\phi_0 = \pi/2$. The black line shows the Ansatz of equation 13 with $\sigma^2 = d_{\text{step}}^2 t/2$ and $|n|_{\max} = 500$. The orange dashed line indicates only the $n = 0$ term and is seen to diverge from both the numerical average and the black line only at late times.

## Statistics

Using the evolution of the noise on the sphere, we can consider the objective collapse dynamics of Eq. (9). For $\tau \ll 1/(JN)$, the noise fluctuates much faster than the state evolves. The state evolution then effectively experiences the average value of the noise, which is always located at the equator, $\eta = \pi/2$. In the limit of infinitesimal $\tau$, this causes the probability $P_{|0\rangle}$ of the state evolving to the fixed point given by the pointer state $|0\rangle$, to become a step-function: $P_{|0\rangle} = \Theta(\theta_0 - \pi/2)$.

We can counteract the tendency towards a step-function distribution of outcomes by introducing a scaling factor $B$ for the ratio of between the coupling strength to the stochastic noise and the non-linear coupling driving the collapse dynamics:

$$\dot{\theta} = -JN \sin(\theta)(\cos(\theta) - B\lambda_1). \tag{14}$$

For large values $B \gg 1$, the stochastic term dominates the collapse process, and the non-linear term can be neglected. This yields a flat probability distribution $P_{|0\rangle} = 1/2$ for the collapse outcome as, expected from earlier work [24].

The probability distribution corresponding to Born's rule, $P_{|0\rangle} = |\cos(\theta_0/2)|^2$, interpolates between the step-function obtained for small values of $\tau$ at $B = 1$, and the constant obtained at large $B$. The probability distributions for various intermediate values of $B$ are shown in Fig. 3, and are seen not to have an inflection point apart from the central point at $\theta_0 = \pi/2$. For any value of $\tau$, There is thus guaranteed to be some value of $B$ at which Born's rule is obtained.

Using a bisection method to numerically determine the values of $B$ yielding Born's rule results in Fig. 4. The three lines correspond to different values of $JN$, and each point is the result of a bisection method to find the value of $B$ for fixed $JN\tau$ and $\theta_0$ that yields Born's rule within a 0.0029 margin. For each combination of $JN\tau$ and $B$, 10000 evolutions are averaged to find the corresponding value of $P_{|0\rangle}$. The noise is modelled as a random walk on the unit sphere with steps of arc length 0.05. To minimise the effect of the fixed step size in the evolution of the noise, the noise takes 100 steps before each new step in the evolution of the state.

In terms of the collapse time $\tau_c$ and the size $dt$ of the time step used in simulations, three regimes can be distinguished in Fig. 4, characterised by $\tau_c \ll \tau$, $dt \ll \tau \ll \tau_c$ and $\tau \lesssim dt$. The first is the regime in which the noise is nearly constant during the time it takes the state

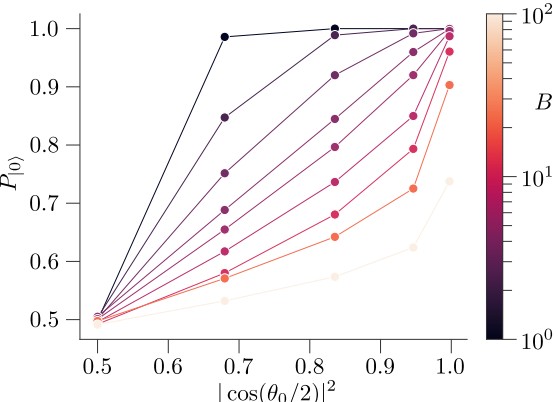

Figure 3: Probability of evolving towards the stable fixed point corresponding to the pointer state $|0\rangle$, as a function of the initial value $\theta_0$. The colours indicate different values for $B$, ranging from 1 to 100. For each instance of the state evolution, 2000 time steps in the evolution of $\theta$ are considered to determine its final state, and each point in the figure corresponds to the final state average of 10000 such evolutions. To minimise the effect of the fixed step size in the evolution of the noise, the noise takes 100 steps of arc length 0.1 before each new step in the evolution of the state.

to collapse. The limiting value of $B$ for large $JN\tau$ matches the analytic result $B = 1$ identified before. For noise processes with any arbitrary but finite correlation time $\tau$, this regime is relevant to macroscopic measurement machines, which collapse instantaneously as $N \to \infty$ (i.e. in the thermodynamic limit).

In the mesoscopic regime, with small but non-zero correlation times $\tau \ll \tau_c$, The relation between $B$ and $JN\tau$ shown in Fig. 4 can be well approximated by the dashed black line over a large range of parameters. The dashed line is a fit of the form $B = \gamma/(JN\tau)^\alpha$, with parameter values found to be $\gamma = 0.92 \pm 0.05$ and $\alpha = 0.50 \pm 0.01$. In this regime we thus find:

$$B^2 \propto 1/(JN\tau) \quad \Longleftrightarrow \quad (BJN)^2 \propto JN/\tau. \tag{15}$$

Here, we write the relation in terms of $JN$, $BJN$, and $1/\tau$, as these are the energy scales defining the dynamics.

Finally, a third regime is visible in Fig. 4, for $\tau < dt$. Here $B$ again becomes independent of $JN\tau$, but does depend on $JN$ in a non-universal way (i.e. not through $J\tau$). This is typical cut-off behaviour arising from the non-commuting limits of $dt \to 0$ and $\tau \to 0$, which indicates that the approximation of a continuous noise process using finite step size breaks down. Decreasing the size of the time steps used in the calculation, or equivalently, considering smaller values of $JN$, results in the shrinking of the third regime. The nonphysical low-$JN\tau$ regime thus vanishes in the $dt \to 0$ limit.

Both in the mesoscopic and in the macroscopic regime, Fig. 4 shows it is possible to employ a Markovian random walk on a unit two-sphere as the stochastic variable in a theory of spontaneous unitarity violation, such that it results in objective collapse obeying Born's rule. For any given correlation time $\tau$ of the external noise, Born's rule is obtained only for a particular relation between the coupling strength $BJN$ of the system to the stochastic noise, and the coupling strength $JN$ to the non-linear term driving the collapse process. The existence of such a relation suggests the stochastic and non-linear processes in models of spontaneous unitarity violation should have a common physical origin. Importantly, it does not imply an assumption of Born's rule in the definition of the dynamics, as the relation between $JBN$ and $JN$ is independent of the state.

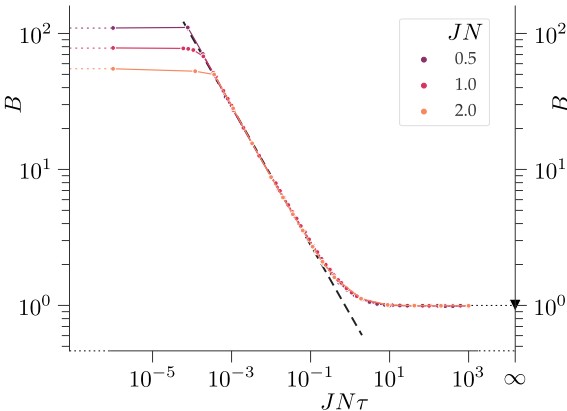

Figure 4: Parameter values yielding Born's rule. The simulation is carried out for $JN = 0.5$ (dark), $JN = 1$ (middle), and $JN = 2$ (light). In each case, three regimes can be identified, corresponding to large, medium and small values of $JN\tau$. The dashed black line indicates a fit for the central regime, of the form $B = \gamma/(JN\tau)^\alpha$ with $\gamma \approx 0.92$ and $\alpha = 0.50$. The downward triangle on the right side indicates the exact result $B = 1$ for the limit $JN\tau \to \infty$, corresponding to constant noise or macroscopic measurement machines. The relation between $JN\tau$ and $B$ can be seen to become non-universal (dependent on $JN$) for small values of $JN\tau$, where the approximation of a continuous noise process using finite step size breaks down. Each circular point along the curves represents the value of $B$ giving Born's rule as determined using a bisection method at fixed $\theta_0$, and probabilities after averaging over 100000 evolutions using arc length steps of 0.05 for the noise.

Notice that starting from Eq. (10) rather than Eq. (9), the stochastic terms appears inside a periodic function. Introducing a parameter $B$ multiplying $\lambda_2$ may then affect the direction of the state evolution, but not its speed. In that situation it is therefore not possible to balance two limiting behaviours as we did in Fig. 3, and Born's rule cannot be obtained at any finite $\tau$. Because the values of the stochastic term in consecutive experiments should not be correlated, this effectively rules out any dynamics based on Eq. (10) models for objective collapse. We thus find that the only consistent form for theories of spontaneous unitarity violation, as applied to an initial two-state superposition, is given by Eq. (14), with the relation between $B$ and $JN\tau$ according to Fig. 4.

**Random walks on other manifolds**

The analysis above can be repeated for random walks on any closed manifold. In general, Eq. (14) yields the limiting behaviours for small $\tau/\tau_c$ and large $B$ shown in Fig. 4. Interpolating between these, a relation between $B$ and $JN\tau$ resulting in Born's rule can in principle be obtained even in the general case. The resulting dynamical equations, however, do not generally allow for a physical interpretation.

To see this, consider the example of a random walk on a two-torus. As shown in detail in the Supplemental Material, the requirement that Born's rule is obtained in the macroscopic regime result in a relation of the form $\lambda_1 \propto \phi$, with $\phi$ one of the two angles parameterizing positions on the two-torus. Although this is a valid algebraic expression, the angle $\phi$ appearing outside of any trigonometric function would render $\lambda_1$ a multi-valued function of $\phi$, which cannot be given a physical interpretation. This feature generally appears for random walks on manifolds with non-zero genus, so that Eq. (14) provides a physical model for objective collapse only if the stochastic variable $\lambda_1$ is modelled by a random walk on a $(d > 1)$-dimensional sphere.

The examples of random walks on the unit 3-sphere and 4-sphere, are worked out in detail in the Supplemental Material. Both yield physical models, which reproduce Born's rule for general $JN\tau$, given a specific relation between $B$ and $JN\tau$. In both cases, the macroscopic, large-$JN\tau$ limit yields $B = 1$, as in the case of the random walk on a two-sphere considered before. In the regime of $\tau$ non-zero but small compared to $\tau_c$, both higher-dimensional cases again yield a relation of the form $B \approx \gamma/(JN\tau)^{0.5}$, but with parameter values $\gamma = 0.936$ (3-sphere) and $\gamma = 0.966$ (4-sphere).

We thus find that a model for spontaneous unitarity violation starting from a superposition over two states must necessarily be of the form of Eq. (14), with the stochastic variable $\lambda_1$ modelled by a random walk on a sphere. For a sphere in any dimension larger than one, and for any non-zero correlation time of the random walk, Born's rule is recovered given a specific relation between $BJN$ and $JN$. The functional form of the required relation depends on the dimension of the sphere only in a parametric fashion.

## 5 Microscopic regime

In the microscopic limit, finally, the measurement machine itself is a quantum system that consists of only a small number of constituent particles (argued for example in Ref. [1] to be fewer than about $10^6$ atoms). In this regime, the dynamics induced by Eq. (14) should be negligible in order to reproduce the well-established adherence of microscopic physics to Schródinger's dynamics. In direct analogy to ny usual type of spontaneous symmetry breaking [36], this implies that the strength of the unitarity-breaking perturbation $J$ must be sufficiently weak to have unobservable effect on any experimentally achievable time scale. Only the regular dynamics governed by Schrödinger's equation then remains, reproducing the experimental fact that the evolution of microscopic objects is well-described by unitary quantum mechanics.

## 6 Discussion

In summary, we considered models of spontaneous unitarity violation, in which a weak non-unitary perturbation of Schrödinger's equation causes the objective collapse of macroscopic quantum systems, while leaving the evolution of microscopic particles unaffected. We restricted attention to models with both a non-unitary, non-linear, deterministic term, and separately a non-unitary, linear, stochastic term. The separation between non-linear and stochastic terms ensures that any noise process representing the evolution of the stochastic variable is independent of the state being collapsed.

Imposing the constraint that Born's rule must be obtained for the final state statistics of the collapse dynamics, we find that there is only one, *unique* form for the evolution starting from a two-state superposition:

$$\dot{\theta} = -JN\sin(\theta)(\cos(\theta) - B\lambda). \tag{16}$$

Here, $\theta$ is an Euler angle parameterising the Bloch sphere, $N$ represents the size (number of particles, mass, or volume) of the collapsing system, while $J$ and $BJ$ are the coupling constants for the non-linear and stochastic processes driving the collapse dynamics. The random variable $\lambda$ is defined on a bounded domain that we take to be $[-1, 1]$, and has a correlation time $\tau$ that we assume to be non-zero.

With these definitions, Born's rule is recovered in the limit of $\tau$ large compared to the collapse time $\tau_c$ if $B = 1$ and the equilibrium probability distribution for $\lambda$ is flat on the interval $[-1, 1]$. For general values of $\tau$, finding Born's rule requires the existence of a relation between

the coupling strength of the stochastic term, $BJN$, and that of the non-linear term, $JN$. For $\tau \ll \tau_c$ the relation is of the form $(BJN)^2 \propto JN/\tau$, independent of the state undergoing collapse (i.e. regardless of the initial state being measured). For intermediate values of $\tau$, the required relation between $BJN$ and $JN$ follows a smooth curve interpolating between the forms at short and long correlation times, as shown in Fig. 4. Importantly, the requirement that there exists a specific relation between coupling strengths suggests a common physical origin for the stochastic and non-linear processes driving the objective collapse process, akin for example, to the relation between drift and dissipation in Einstein's description of Brownian motion [37, 38].

Finally, modelling the stochastic process as an unbiased random walk, the requirement of obtaining Born's rule restricts the possible types of manifold on which the random walk takes place. It needs to be closed for the equilibrium distribution of $\lambda$ cover a bounded interval, it should have genus zero to allow for a single-valued map between $\lambda$ and a coordinate on the manifold, and it should have dimension two or larger to allow for the equilibrium distribution of of $\lambda$ to be flat. These conditions limit the possible types of physical processes that can provide the stochastic process driving collapse dynamics. Together with the identification of a physical relation between stochastic and non-linear processes, the results presented here thus constrain and point the way towards a fully microscopic theory underlying objective collapse models based on spontaneous unitarity violation.

# Acknowledgments

We thank Ulrike Nitzsche for her technical assistance.

# A Appendix

This appendix details the calculation of probability distribution functions for random walks on closed manifolds used in the main text.

## A.1 Constant noise limit on the two-sphere

Consider a two-sphere parameterised by the latitude and azimuthal angles $\eta$ and $\phi$. Combining these angles into a vector $\vec{g} = (\eta, \phi)$, any alternative set of coordinates can be written in terms of a vector $\vec{f}(\vec{g})$, whose components are functions of $\eta$ and $\phi$. We define:

$$E = \partial_\eta \vec{f} \cdot \partial_\eta \vec{f} \,,$$
$$F = \partial_\eta \vec{f} \cdot \partial_\phi \vec{f} \,,$$
$$G = \partial_\phi \vec{f} \cdot \partial_\phi \vec{f} \,.$$

Under the coordinate transformation from $\vec{g}$ to $\vec{f}$, the probability distribution function defined on the two-sphere transform as [35]:

$$P_B(\vec{f}) = \frac{P_A(\vec{g})}{\sqrt{EG - F^2}} \,. \tag{A.1}$$

Here, $\sqrt{EG - F^2}$ denotes the Jacobian of the coordinate transformation, and the subscripts $A$ and $B$ denote that the probability distribution is expressed in original and transformed coordinates respectively.

Considering an unbiased random walk on the two-sphere, we know that the infinite-time probability distribution in terms of Cartesian coordinates is $P_B(x, y, z) = 1/(4\pi)$. That is, under the Markovian random process every point on the manifold obtained with equal likelihood in the equilibrium distribution. Applying the prescription of Eq. (A.1) The distribution $P_A(\eta, \phi)$ in terms of Euler angles is found to obey:

$$\frac{1}{4\pi} = \frac{P_A(\eta, \phi)}{|\sin(\eta)|}. \tag{A.2}$$

Defining probability distribution functions for the individual Euler angles in terms of independent sampling, $P_A(\eta, \phi) = P_A(\eta)p_A(\phi)$, the distribution of $\eta$ follows from:

$$P_A(\eta) \int_0^{2\pi} P_A(\eta', \phi)|_\eta d\phi = \frac{\sin(\eta)}{4\pi} \int_0^{2\pi} d\phi$$

$$\Rightarrow \quad P_A(\eta) = \frac{1}{2} \sin(\eta).$$

### A.2 Fluctuating noise on the two-sphere

Starting from the knowledge that at short times, the probability distribution obtained in a random walk on the sphere starting from the point $(\eta_0, \phi_0)$ is Gaussian in the arc distance travelled, we can write:

$$P_A(\eta) \approx \int_0^{2\pi} d\phi \bar{N} \sin(\eta) \exp\left(\frac{-1}{2\sigma^2}\Delta^2\right). \tag{A.3}$$

Here, $\Delta = \arccos[\sin(\eta_0)\sin(\eta)\cos(\phi_0 - \phi) + \cos(\eta_0)\cos(\eta)]$ is the arc distance between the initial point $(\eta_0, \phi_0)$ and the point $(\eta, \phi)$, while $\bar{N}$ denotes a normalisation factor. The analytical result of this integral is not known. However, for $x \ll 1$ we can write:

$$\arccos(x)^2 = \pi^2/3 - \pi x + O(x^2).$$

Cutting off the series at this order corresponds to taking the short time limit. In that limit, we can solve the integral using:

$$\int_0^{2\pi} \exp(b\cos(\phi) + c\sin(\phi))d\phi = 2\pi I_0\left(\sqrt{b^2 + c^2}\right).$$

Here, $I_0$ denotes the Bessel function. We are then left with an approximate form for the probability distribution function valid at short times:

$$P_A(\eta) \approx \bar{N} \sin(\eta) I_0\left(\frac{\sin(\eta_0)\sin(\eta)}{2\sigma^2}\right) e^{\frac{\cos(\eta_0)\cos(\eta)}{2\sigma^2}}.$$

Here, all prefactors are absorbed into the normalisation $\bar{N}$. This expression is an approximation of the actual distribution function because arccos is a multi-valued function, and we only consider the domain $[0, \pi]$, thus ignoring the tails of the Gaussian distribution extending all the way around the sphere. To correct for this omission, we can include all domains of the arccos:

$$P_A(\eta) = \sum_{n=-\infty}^{\infty} \int_0^{2\pi} d\phi \bar{N} \sin(\eta) \exp\left(\frac{-1}{2\sigma^2}(\arccos[\sin(\eta_0)\sin(\eta)\cos(\phi_0 - \phi)\right.$$
$$\left. + \cos(\eta_0)\cos(\eta)] + 2n\pi)^2\right). \tag{A.4}$$

In the infinite sum over $n$, all terms evaluate to weighted Bessel functions.

### A.3 Random walk on the two-torus

As an example of a random walk on a bounded two-dimensional manifold other than a sphere, consider a torus described by the Cartesian coordinates:

$$x(\eta, \phi) = (R + r\cos(\eta))\cos(\phi),$$
$$y(\eta, \phi) = (R + r\cos(\eta))\sin(\phi),$$
$$z(\eta, \phi) = r\sin(\eta).$$

Here, the angles $\eta$ and $\phi$ are both defined on the interval $[0, 2\pi)$. The angle $\phi$ denotes the rotation around the axis of revolution of the torus, while $\eta$ is the angle describing rotations around the surface of the torus at fixed $\phi$. The constant $r$ denotes the radius of the circle whose revolution yields the torus, while $R$ is the distance between the centre of the torus to the middle of the circle with radius $r$. The Jacobian for the transformation between Cartesian coordinates and the coordinates $(\eta, \phi)$, is given by $|J| = rR + r^2\cos(\eta)$.

If the probability distribution function $\rho(x, y, z)$ has equal value for every valid combination of $x$, $y$, and $z$, then we can deduce the probability distributions for the individual angles $\eta$ and $\phi$ as before:

$$P_A(\eta) = \frac{R + r\cos(\eta)}{2\pi R}, \qquad P_A(\phi) = 1/(2\pi).$$

To obtain Born's rule in the constant noise limit, we must identify a coordinate $\lambda = g(\eta, \phi)$ on the two-sphere, such that its probability distribution function becomes either flat in the domain $[-1, 1]$ (for the collapse process with $\lambda_1$) or equal to $1/2\sin(\lambda)$ in the domain $[0, \pi]$ (for the $\lambda_2$ process). The only possible ways of realising these constraints are given by:

$$\lambda_1(\phi) = \phi/\pi - 1,$$
$$\text{or } \lambda_1(\eta) = \eta/\pi + \frac{r}{\pi R}\sin(\eta) - 1,$$

$$\lambda_2(\phi) = \arccos(1 - \phi/\pi),$$
$$\text{or } \lambda_2(\eta) = \arccos\left(\frac{\eta}{\pi} + \frac{r}{\pi R}\sin(\eta) - 1\right).$$

Notice that the angles $\eta$ and $\phi$ appearing outside of any trigonometric functions render $\lambda_1$ a multi-valued function of $\phi$, while the arccos has the same effect on the function $\cos(\theta - \lambda_2)$ appearing in the state dynamics. These functions therefore cannot be given a physical interpretation.

### A.4 Higher-dimensional manifolds

On higher-dimensional manifolds, the procedure for obtaining the probability distribution function for a single component is a straightforward generalisation of the procedure on the two-sphere. Using again the assumptions of independent sampling and equal likelihood for obtaining any point on the manifold in the infinite-time limit, we can write:

$$P_A(\eta) = J(\eta, \{\Xi\})/V. \tag{A.5}$$

Here, $V$ is the volume of the $d$-dimensional manifold and $J$ the Jacobian of the transformations from Cartesian coordinates to the coordinates $(\eta, \{\Xi\})$, with $\{\Xi\}$ a list of $d-1$ angles [35].

To obtain Born's rule in the constant noise limit, we again need to define coordinate $\lambda_1$ or $\lambda_2$, such that $P(\lambda_1) = 1/2$ or $P(\lambda_2) = 1/2\sin(\eta)$. Formally following the same steps as before

this yields the possible definitions:

$$\lambda_1(\eta) = \int_a^\eta d\eta' \frac{\sin(\eta')}{2J(\eta', \{\Xi\})},$$

$$\lambda_2(\eta) = \int_a^\eta d\eta' \frac{1}{2J(\eta', \{\Xi\})}.$$

Here, the constant $a$ will be determined by the domain of $\lambda$. The equations can be evaluated for any choice of coordinates on the manifold.

### A.4.1 Three-sphere

Directly applying this procedure on the unit three-sphere, we have the relation between Cartesian coordinates and Euler angles given by:

$$x_1 = \cos(\phi_1),$$
$$x_2 = \sin(\phi_1)\cos(\phi_2),$$
$$x_3 = \sin(\phi_1)\sin(\phi_2)\cos(\phi_3),$$
$$x_4 = \sin(\phi_1)\sin(\phi_2)\sin(\phi_3).$$

This implies the Jacobian and transformed probability distributions:

$$|J_4| = \sin^2(\phi_1)\sin(\phi_2),$$
$$\frac{1}{2\pi^2} = \frac{P_A(\phi_1, \phi_2, \phi_3)}{\sin^2(\phi_1)\sin(\phi_2)},$$
$$P_A(\phi_1, \phi_2) = \frac{1}{\pi}\sin^2(\phi_1)\sin(\phi_2).$$

Using these, we find the probability distributions for individual coordinates:

$$P_A(\phi_3) = \frac{1}{2\pi},$$
$$P_A(\phi_2) = \frac{1}{2}\sin(\phi_2),$$
$$P_A(\phi_1) = \frac{2}{\pi}\sin^2(\phi_1).$$

These, finally allow a definition for the functions featuring in the state dynamics such that Born's rule is obtained in the constant noise limit:

$$\lambda_1(\phi_1) \propto (\eta - \sin(\eta))\cos(\eta),$$
$$\text{or } \lambda_1(\phi_2) = \cos(\phi_2).$$

The first possibility can again not be given a physical interpretation due to the appearance of the angle $\eta$ outside of a trigonometric function. We thus restrict attention to the definition $\lambda_1(\phi_2) = \cos(\phi_2)$ from here on.

Having found dynamics that yields Born's rule in the static noise limit, we can consider time-varying noise by directly simulating a random walk on the three-sphere, as we did in the main text for the two-sphere. Again, we find that there is a specific value for the parameter $B$ at any value of $JN\tau$ which yields Born's rule, as shown by the red line in Fig. 5. As for the two-sphere, the low-$JN\tau$ behaviour of $B$ can be fitted with a function of the form $B = \gamma/(JN\tau)^\alpha$, which in the case of the three-sphere yields in the best-fit parameters $\gamma = 0.936$ and $\alpha = 0.49$.

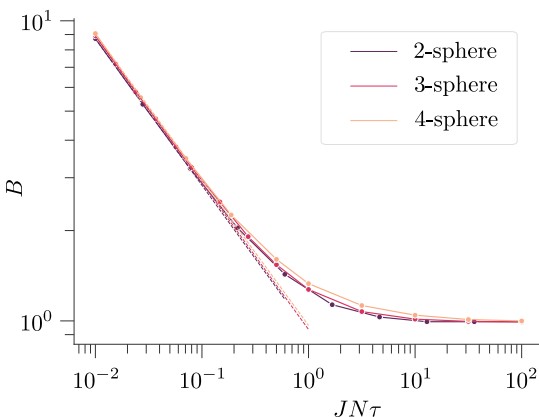

Figure 5: The relation between $B$ and $JN\tau$ yielding Born's rule when the stochastic process is interpreted as a random walk on either a 2-, 3-, or 4-sphere. The dashed lines show fits of the low-$JN\tau$ regime of the form $B = \gamma/(JN\tau)^\alpha$. The best-fit values for the 2-sphere are $\gamma = 0.919$ and $\alpha = 0.49$, while those for the 3-sphere are $\gamma = 0.936$ and $\alpha = 0.49$, and we find $\gamma = 0.966$ and $\alpha = 0.49$ for the 4-sphere.

### A.4.2 Four-sphere

Again applying the same procedure to the unit four-sphere, the relation between Cartesian coordinates and Euler angles is given by:

$$
\begin{aligned}
x_1 &= \cos(\phi_1), \\
x_2 &= \sin(\phi_1)\cos(\phi_2), \\
x_3 &= \sin(\phi_1)\sin(\phi_2)\cos(\phi_3), \\
x_4 &= \sin(\phi_1)\sin(\phi_2)\sin(\phi_3)\cos(\phi_4), \\
x_5 &= \sin(\phi_1)\sin(\phi_2)\sin(\phi_3)\sin(\phi_4).
\end{aligned}
$$

This implies the Jacobian and transformed probability distributions:

$$
\begin{aligned}
|J_4| &= \sin^3(\phi_1)\sin^2(\phi_2)\sin(\phi_3), \\
\frac{1}{2\pi^2} &= \frac{P_A(\phi_1, \phi_2, \phi_3, \phi_4)}{\sin^3(\phi_1)\sin^2(\phi_2)\sin(\phi_3)}, \\
P_A(\phi_1, \phi_2, \phi_3) &= \frac{1}{\pi}\sin^3(\phi_1)\sin^2(\phi_2)\sin(\phi_3).
\end{aligned}
$$

Using these, we find the probability distributions for individual coordinates:

$$
\begin{aligned}
P_A(\phi_4) &= \frac{1}{2\pi}, \\
P_A(\phi_3) &= \frac{1}{2}\sin(\phi_3), \\
P_A(\phi_2) &= \frac{2}{\pi}\sin^2(\phi_2), \\
P_A(\phi_1) &= \frac{3}{4}\sin^3(\phi_1).
\end{aligned}
$$

These, finally allow a definition for the functions featuring in the state dynamics such that Born's rule is obtained in the constant noise limit:

$$
\lambda_1(\phi_2) = \cos(\phi_3).
$$

Repeating the random walk simulation, but on the four-sphere we obtain the relation between $B$ and $JN\tau$ indicated by the orange line in Fig. 5. The relation at small $JN\tau$ can be fit with a function of the same form and nearly identical best-fit parameter values as in the case of the two-sphere and three-sphere.

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
