# Peer review of "Stochastic field dynamics in models of spontaneous unitarity violation"

_SciPost Physics Core, doi:SciPost Phys. Core 7, 012 (2024)_

## Round 1 · Referee Report · Anonymous (Referee 1) · 2024-2-9

Strengths

clear predictions

Weaknesses

esoteric topic

Report

Report on the paper “Stochastic Field Dynamics and Models of Spontaneous Unitary Violation”.

The manuscript represents a discussion of objective collapse theories as a solution to the quantum measurement problem. I'm not an expert in the field. But I still find the paper very well written and largely understandable, even to a non-expert. While I do have hile I do have several suggestions for possible improvements to the paper, I nevertheless think that the paper after these improvements have been made Is certainly publishable in SciPost.

The main attractive point of the present paper, and a point which distinguishes it from many other papers on fundamental theories, is that it can actually rule out equation 10 or equation 3 as a viable model of an objective collapse theory.

Requested changes

  1. If I understand it correctly. Then the idea of objective collapse theories is to model the collapse of the wave function by a stochastic non-unitary term. Augmented by a non unitary nonlinear element. This is mentioned in the abstract and also discussed on the first page and first paragraph of the right hand column. Perhaps this could be discussed a little bit further and made a little bit clearer to the non-expert.

2.. The use of equations 2 and three should be better motivated than simply saying. These are the 2. Equations possible. A few words regarding why this is physically required or plausible would be helpful.

  1. I'm slightly at a loss how exactly the requirement to adhere to Born’s rule enters in the discussion here. In particular, it is never stated what you actually have in mind regarding Born's rule and how it connects to equations 2 and 3?

  2. At the beginning of Section 4 mesoscopic regime we make a connection between a violation of Bournes rule and faster than light communication. I do not see this connection. Please explain.

  3. You introduce the factor B as the ratio between the coupling strengths of the stochastic noise and the nonlinear coupling. Later on you discuss this as JN relative to BJN. Why not simply use B as the relative strength? I found this confusing.

  4. In the letter parts of the paper you discuss equations 9 and 10 only. But these are indeed just modified equations 2 and three. I think it would be better if you were to make a connection back to your original equations 2 and three at least. Remind the reader once in a while that it is not equations 9 and 10 that you're looking at, but really equations 2 and three.

  5. In Section 5, you argue that constant J has to be weak. It would be good to have some estimate of what “weak” actually means in this context. For example, something that relates it to the strengths of \hbar or some other such fundamental constant. If that is not possible, then it would also be good to highlight why this is not possible.

  6. The paper certainly represents a serious effort to discuss objective collapse theories, and I'm delighted by the many appendices that give further detail.

I have found only very few typos. But there are some. So the authors might want to have a further check.

---

## Round 2 · Author Response

We are grateful to the referee for their positive evaluation of our manuscript, and for their constructive remarks.

Below, we reply to the individual points raised in the referee report:

1) We thank the referee for the suggestion and are happy to add additional discussion of this point in the revised manuscript. 2) In the revised manuscript, we added the brief explanation proposed by the referee. 3) We thank the referee for pointing out the lack of clarity here, and add a more detailed discussion of the role of Born's rule to the revised manuscript. 4) As suggested, we add an explanation of how violations of Born's rule allow for faster than light communication. 5) We thank the referee for the suggestion, but think this is a matter of taste. Since BJN and JN are the actual energy scales defining the time evolution operator, we prefer to use these. 6) We agree with the referee that equations 9 and 10 are equivalent to equations 2 and 3 after implementation of consistency constraints on the parameters lambda and a. We emphasize this in the revised manuscript. 7) We thank the referee for the question. "Weak" in this case has the same meaning as in the usual theory of spontaneous symmetry breaking: it refers to anything beyond the limit of practical control. We add a brief explanation of this point in the revised manuscript. 8) We thank the referee for their kind words. 9) We checked the revised manuscript for spelling errors.

Once again, we thank the referee for their helpful input and positive evaluation.

---

## Round 2 · List of Changes

See above

---

## Round 3 · Referee Report · Anonymous (Referee 1) · 2024-2-25

Report

Really, again? I presume you can reuse my old comments ;-)

Requested changes

There is a typo tau -> \tau after Eq. (15).

Thanks for making those changes and I am happy to accept as is now.

---

## Round 3 · Author Response

In the attached revised manuscript, we implemented the additional suggestion of the reviewer, applied the SciPost stylefile, and included doi in the bibliography.

As requested, please also find attached the difftex file, comparing the text of the current version (before updating the style and bibliography) to that of the original submission.

NB: Since SciPost does not allow uploading multiple files upon resubmission, the diff-file is included in the end of the revised manuscript.

---

## Editorial Decision

published